# LEARNING USER PREFERENCES FOR IMAGE GENERATION MODELS

## ABSTRACT

User preference prediction requires a comprehensive and accurate understanding of individual tastes. This includes both surface-level attributes, such as color and style, and deeper content-related aspects, such as themes and composition. However, existing methods typically rely on general human preferences or assume static user profiles, often neglecting individual variability and the dynamic, multifaceted nature of personal taste. To address these limitations, we propose an approach built upon Multimodal Large Language Models, introducing contrastive preference loss and preference tokens to learn personalized user preferences from historical interactions. The contrastive preference loss is designed to effectively distinguish between user "likes" and "dislikes", while the learnable preference tokens capture shared interest representations among existing users, enabling the model to activate group-specific preferences and enhance consistency across similar users. Extensive experiments demonstrate our model outperforms other methods in preference prediction accuracy, effectively identifying users with similar aesthetic inclinations and providing more precise guidance for generating images that align with individual tastes.

## 1 INTRODUCTION

Recent work in generative models (Ho et al., 2020; Dhariwal & Nichol, 2021; Sohl-Dickstein et al., 2015; Nichol et al., 2022; Saharia et al., 2022; Rombach et al., 2022; Ren et al., 2024; Esser et al., 2024; Sauer et al., 2024a; Mo et al., 2025; Zhou et al., 2025; Zhang et al., 2025; Ba et al., 2025) has significantly advanced the field of image generation. However, these models often produce generic outputs that may not align with the diverse and nuanced preferences of each individual user. A particularly promising direction within this domain is user preference prediction based on generated images, which has garnered increasing attention due to its capability to guide generative models tailored to individual preferences. By aligning generated content with specific user interests, this direction holds the potential to deliver unique user experiences, thereby enhancing user satisfaction and engagement.

The feasibility of such personalized approaches is supported by psychological research, which suggests that aesthetic preference is not arbitrary but often reflects a mixture of low-level visual features (e.g., color, contrast) and high-level semantic content (e.g., subject matter, composition) (Iigaya et al., 2021). Such findings support the assumption that individual taste can be inferred from observable image properties, laying the foundation for data-driven modeling of personalized visual preference.

Building on this foundation, the task of user preference prediction becomes well-defined: given reference data, typically a set of liked and disliked images, the task of user preference prediction is to identify preferences, such as color and content, that align with a user's tastes. Fig. 1 provides an illustrative example. Existing preference prediction models such as PickScore (Kirstain et al., 2023), ImageReward (Xu et al., 2023), and HPS (Wu et al., 2023b;a) evaluate human preferences at a general level, without granular individual-specific adaptation. Moreover, recent individual-level personalized preference modeling (Salehi et al., 2024; Shen et al., 2024) presents three primary issues: (1) focus on superficial attributes like color and style, which limits their ability to capture the essence of a deep content-level preference and (2) overlook the significance of users' disliked images, which provide valuable defeatist feedback and relative preference signals for refining pref-

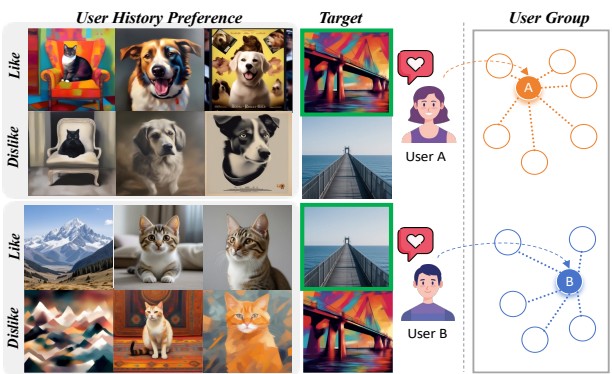

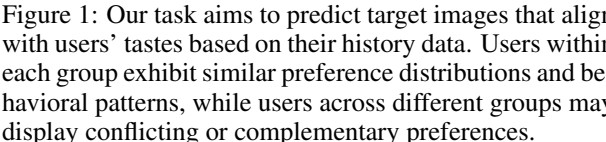

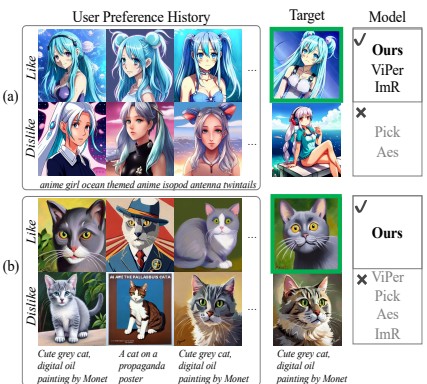

Figure 1: Our task aims to predict target images that align with users' tastes based on their history data. Users within each group exhibit similar preference distributions and behavioral patterns, while users across different groups may display conflicting or complementary preferences.

Figure 2: Qualitative comparison of user preference prediction. (a) and (b) illustrate user-specific preferences for style and content, respectively. The green boxes represent the desired outputs that match the user's preferences.

erence understanding, (3) fail to utilize the fact that users with similar tastes might share preferences for certain types of images.

Learning user preferences, while ostensibly analyzing individual historical reference data, fundamentally requires global modeling—capturing both inter-user divergence, which defines individualized content needs, and cross-user commonality, which enables structured learning across similar users. While prior works in text-to-image generation often treat users as independent units, we posit that users frequently exhibit shared preference patterns. This motivates us to formulate a user group structure, where intra-group consistency and inter-group divergence guide the preference modeling process. Our approach is also inspired by recent developments in recommendation systems, where contrastive clustering has been successfully employed to learn group-level behavior representations (Lan et al., 2024). Analogously, we design a multimodal preference learning framework built upon Multimodal Large Language Models (MLLMs) (Laurençon et al., 2023; 2024; Hu et al., 2024; Yao et al., 2024; Li et al., 2024; Liu et al., 2024b;a; An et al., 2025), in which we introduce contrastive preference loss terms to sharpen decision boundaries, and incorporate learnable preference tokens to dynamically encode cluster-specific attributes. This design not only enhances individual preference discrimination but also promotes alignment within user groups, leading to more consistent and structured preference modeling. Our contributions are summarized as follows:

- We introduce a MLLM-based contrastive learning framework that enables the model to learn discriminative features from users' liked and disliked data, effectively capturing fine-grained user preferences by modeling relative preference relationships among samples.

- We leverage learnable preference tokens to capture shared interests among users, allowing the model to generalize better across users with similar tastes.

- Experimental results demonstrate that our model outperforms existing methods in preference recognition accuracy. It is able to identify users with similar tastes and effectively generalizes to new users with similar preferences. Furthermore, it provides more precise guidance for generating personalized content.

## 2 RELATED WORK

Modeling user preferences in text-to-image generation is essential for improving alignment with human aesthetics and expectations. Existing research in this area can be broadly categorized into two main categories: general preference modeling, which focuses on capturing collective human judgments to enhance overall image quality, and user-specific preference modeling, which personalizes image generation based on individual tastes and behaviors.

Figure 3: Overview of our MLLM-based preference learning framework. (a) The visual encoder and text embedding module extract preference representations $x_u^{+/-}$ by processing the preference history $\mathcal{S}$ and a target item $z_{\text{pos/neg}}$. (b) The framework is trained using a base loss $L_{\text{base}}$ to predict preference labels, and a contrastive preference loss $L_{\text{CP}}$ that enhances separability between liked and disliked items. Additionally, learnable preference tokens $P_v$ are introduced to model shared user interests.

**General Preference Modeling for Human-Aligned Image Generation.** Researchers have explored various strategies to improve alignment, categorized into three approaches: (1) Filtering Training Data with Preference Scores. By selecting training data based on human feedback scores or automated metrics, models can benefit from high-quality examples that reflect specific user demands. For instance, Liang *et al.* (Liang et al., 2024a) demonstrates how filtering data based on feedback scores leads to improved model performance, as it ensures that only the most relevant examples are used for fine-tuning. Similarly, HPS (Wu et al., 2023b;a) builds upon this concept by introducing a scoring mechanism to prioritize image-text pairs closely aligned with user preferences, making the model more responsive to varied user expectations. (2) Reward-Weighted Fine-Tuning for Human-Aligned Models. In this approach, models are fine-tuned using reward signals that weigh heavily on user satisfaction. Lee *et al.* (Lee et al., 2023) exemplifies this by incorporating feedback-based rewards during training, which generates outputs aligned with user preferences. Furthermore, ImageReward (Xu et al., 2023) provides a structured method for translating human judgments into reward functions, which guides the model's fine-tuning process. By giving greater importance to rewards that capture user satisfaction, these methods tailor the model's outputs to reflect diverse and nuanced user tastes. (3) Reinforcement Learning for Preference Optimization (Mo et al., 2024; Hao et al., 2023; Chen et al., 2024; Liang et al., 2024b). Recent work (Mo et al., 2024; Hao et al., 2023) uses reinforcement learning to optimize the input prompts for high-quality images. DiffusionDPO (Wallace et al., 2024) leverages user preferences to fine-tune the model, improving its ability to generate images that reflect user choices. D3PO (Yang et al., 2024) eliminates the need to train an explicit reward model by directly fine-tuning the diffusion model using human preference data. Its training strategy is grounded in human preference comparisons and achieves performance comparable to traditional reward-based methods.

**User-Specific Preference Modeling and Personalized Image Generation.** In recent advancements in personalized image generation, several approaches have emerged to better align generative models with individual needs. While customization-based methods like DreamBooth (Ruiz et al., 2023) and Textual Inversion (Gal et al., 2023) focus on incorporating specific objects or styles through fine-tuning with a few example images, user-preferred personalized image generation takes a different approach by learning broader user preferences and aesthetic tendencies. These approaches, while effective for small datasets, focus on integrating specific instances rather than broader user behaviors. To improve personalization, Salehi *et al.* (Salehi et al., 2024) proposes a standardized process to collect user preferences using a few query images. User feedback is then systematically incorporated to adjust the preferences extracted from the user during the generation process. Additionally, Shen *et al.* (Shen et al., 2024) introduces a method to integrate user-specific preferences across different modalities, such as text and images, creating personalized outputs by leveraging historical interactions, such as clicks and conversations. This multimodal approach significantly enhances the models' adaptability to align with user needs.

## 3 METHOD

Our approach develops a discriminative preference model that aligns with user-specific tastes. We leverage each user's preference history $\mathcal{S} = \{(I_{\text{pos}}, I_{\text{neg}}, T)_i\}_{i=1}^{N_{\text{ref}}}$ containing $N_{\text{ref}}$ liked/disliked im-

ages for prompt $T$. For any target image pair $(z_1, z_2)$, we define $D_u(z_1, z_2) = \mathbf{1}[Q(\mathcal{S}_u, z_1) > Q(\mathcal{S}_u, z_2)]$, which equals 1 if user $u$ prefers $z_1$ over $z_2$ and 0 otherwise, where $Q(\mathcal{S}, z)$ is a preference scoring function.

Our approach aims to achieve global modeling of user preferences. Therefore, we formalize the user preference structure through the following assumption:

---

**Assumption 1.** *(User Preference Group Structure)*. *We assume users partition into $K$ groups $\{\mathcal{U}_k\}_{k=1}^K$ satisfying:*
**Intra-group homogeneity:** *For users $i, j \in \mathcal{U}_k$:*

$$d(\mathcal{S}_i, \mathcal{S}_j) \leq \rho_k, \mathbb{E}\left[|Q(\mathcal{S}_i, z) - Q(\mathcal{S}_j, z)|\right] \leq \epsilon_k \tag{1}$$

$$\mathbb{P}\left[D_i(z_1, z_2) = D_j(z_1, z_2)\right] \geq 1 - \alpha_k \tag{2}$$

**Inter-group heterogeneity:** *For users $i \in \mathcal{U}_k, j \in \mathcal{U}_{l \neq k}$:*

$$d(\mathcal{S}_i, \mathcal{S}_j) \geq \delta_{kl}, \mathbb{E}\left[|Q(\mathcal{S}_i, z) - Q(\mathcal{S}_j, z)|\right] \geq \max(\epsilon_k, \epsilon_l) \tag{3}$$

$$\mathbb{P}\left[D_i(z_1, z_2) \neq D_j(z_1, z_2)\right] \geq 1 - \beta_{kl} \tag{4}$$

---

Here, $d(\cdot, \cdot)$ is a distance metric on user preference histories, $\epsilon_k$ controls the similarity of preference scores within group; $1 - \alpha_k$ guarantees the consistency of intra-group decisions; $1 - \beta_{kl}$ ensures the divergence of inter-group decisions. This assumption illustrates that: intra-group homogeneity ensures users within the same group have similar preferences and consistent decisions, while inter-group heterogeneity guarantees significant preference differences and decision discrepancies between different groups.

Motivated by Assumption 1, we propose a multimodal large language model-based contrastive preference learning framework. As shown in Fig. 3, our method learns user preferences through contrastive learning on image pairs and employs learnable preference tokens to encode individual aesthetic patterns, enabling personalized preference modeling.

### 3.1 PREFERENCE LEARNING OBJECTIVE

We denote our model as $\mathcal{M}$, which conditions on a user's preference history $\mathcal{S}$ to assess the likelihood of a user favoring a particular item $z$. For the target item $z$, we define user preference as $z_{\text{pos}}$ if the user likes the item and $z_{\text{neg}}$ if the user dislikes it. We define a comprehensive loss function that combines a base classification loss with a contrastive preference loss, aiming to improve the model's ability to distinguish between "like" and "dislike" predictions.

#### 3.1.1 BASE LOSS.

The base loss, $\mathcal{L}_{\text{base}}$, aims to minimize the classification error across both "like" and "dislike" samples. Let $\mathcal{M}^+(\mathcal{S}, z)$ and $\mathcal{M}^-(\mathcal{S}, z)$ represent the logit outputs for predicting "like" and "dislike" outcomes for a sample $z$, respectively. The associated ground-truth labels are represented as $\mathbf{y}_{\text{pos}}$ and $\mathbf{y}_{\text{neg}}$, respectively. The base loss is defined as:

$$\mathcal{L}_{\text{base}} = \frac{1}{2} \left( \mathcal{L}(\mathcal{M}^+(\mathcal{S}, z_{\text{pos}}), \mathbf{y}_{\text{pos}}) + \mathcal{L}(\mathcal{M}^-(\mathcal{S}, z_{\text{neg}}), \mathbf{y}_{\text{neg}}) \right), \tag{5}$$

where $\mathcal{L}(\cdot)$ denotes a classification loss function.

**Further Enhancing Preference Extraction.** While $\mathcal{L}_{\text{base}}$ capture basic preference patterns, they fail to enforce discriminative separability between liked and disliked items, thus violating the group structure constraints in Assumption 1. Consider the model's preference prediction function:

$$\mathcal{Q}(\mathcal{S}, z) = \frac{\exp(\mathcal{M}^+(\mathcal{S}, z))}{\exp(\mathcal{M}^+(\mathcal{S}, z)) + \exp(\mathcal{M}^-(\mathcal{S}, z))}. \tag{6}$$

Crucially, $L_{\text{base}}$ lacks mechanisms to explicitly maximize the logit margin of "like" and "dislike" predictions. When $\mathcal{M}^+(\mathcal{S}, z) \approx \mathcal{M}^-(\mathcal{S}, z)$, we have $\mathcal{Q}(\mathcal{S}, z) \approx 0.5$, leading to ambiguous decision boundaries. Such predictions fundamentally violate Assumption 1 in two ways: (1) users within the same group $\mathcal{U}_k$ have similar preference scores $Q(\mathcal{S}_i, z_{\text{pos}}) \approx Q(\mathcal{S}_j, z_{\text{pos}}), Q(\mathcal{S}_i, z_{\text{neg}}) \approx Q(\mathcal{S}_j, z_{\text{neg}})$

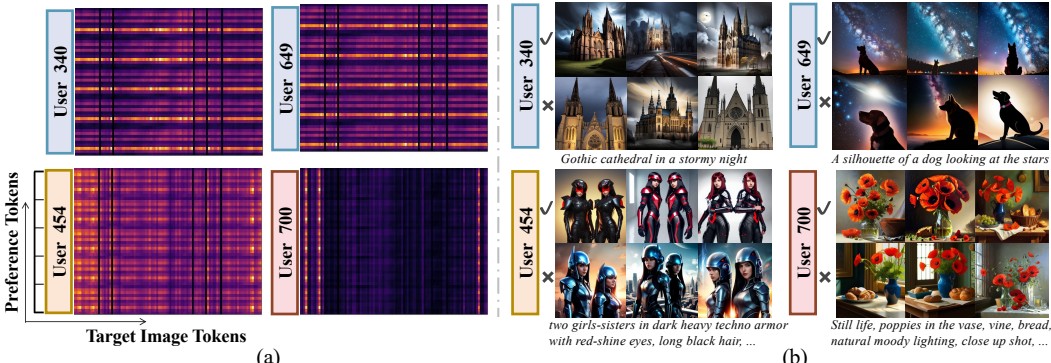

Figure 4: (a) Attention scores $\mathcal{A}$ represent interactions between preference tokens and target image tokens for individual users. Each user has a unique reference history, and we concatenate the same target image to the input sequence across users. For each user, the horizontal axis represents tokens from the target image, while the vertical axis represents the preference tokens. Each user has five random re-orderings of reference images. (b) Examples of images liked (✓) or disliked (✗).

may make inconsistent pairwise decisions, violating the intra-group decision consistency constraint; and (2) when users from different groups $\mathcal{U}_k$ and $\mathcal{U}_l$ both exhibit ambiguous predictions near $0.5$, violating the inter-group score divergence constraint. Simultaneously, their decisions may become unexpectedly similar, thereby violating the inter-group decision divergence requirement.

To further enhance preference extraction, we propose a contrastive preference learning approach that enforces $\mathcal{M}^+(\mathcal{S}, z_{\text{pos}}) \gg \mathcal{M}^+(\mathcal{S}, z_{\text{neg}})$ and $\mathcal{M}^-(\mathcal{S}, z_{\text{neg}}) \gg \mathcal{M}^-(\mathcal{S}, z_{\text{pos}})$. This contrastive mechanism pushes preference scores away from the ambiguous boundary, creating decisive preference predictions with higher confidence. A detailed mathematical analysis is provided in Appendix B.

### 3.1.2 CONTRASTIVE PREFERENCE LOSS

We introduce two contrastive preference loss terms, $\mathcal{L}_+$ and $\mathcal{L}_-$, which enhance the model's ability to differentiate between "like" and "dislike" predictions by emphasizing their relative rankings.

The positive preference loss $\mathcal{L}_+$ ensures the model's positive logits favor positive samples over negative samples. Conversely, the negative preference loss $\mathcal{L}_-$ ensures the model's negative logits favor negative samples over positive samples. Together, these losses push predictions away from ambiguous boundaries:

$$\mathcal{L}_+ = -\frac{1}{N} \sum_{i=1}^{N} \log \sigma(\mathcal{M}^+(\mathcal{S}, z_{\text{pos}}) - \mathcal{M}^+(\mathcal{S}, z_{\text{neg}}))$$

$$\mathcal{L}_- = -\frac{1}{N} \sum_{i=1}^{N} \log \sigma(\mathcal{M}^-(\mathcal{S}, z_{\text{neg}}) - \mathcal{M}^-(\mathcal{S}, z_{\text{pos}}))$$

(7)

where $N$ is the number of samples and $\sigma$ is the sigmoid function. The total contrastive preference loss is the sum of these components, $\mathcal{L}_{\text{CP}} = \mathcal{L}_+ + \mathcal{L}_-$.

The final loss function combines the base loss with the contrastive preference loss to enhance the model's ability to distinguish user preferences: $\mathcal{L}_{\text{all}} = \mathcal{L}_{\text{base}} + \mathcal{L}_{\text{CP}}$. This helps the model optimize for nuanced preference distinctions, leading to more accurate and effective predictions.

### 3.2 LEARNABLE PREFERENCE TOKENS

Based on Assumption 1, we need to adaptively identify user groups and activate corresponding preference patterns without group labels, ensuring intra-group homogeneity and inter-group heterogeneity. Our key insight is leveraging the inherent soft-clustering properties of attention mechanism for personalized group discovery and preference activation.

Consider the core computation of the attention mechanism: $A(\mathcal{Q}, \mathcal{K}) = \text{softmax}\left(\frac{\mathcal{Q}\mathcal{K}^T}{\sqrt{d_k}}\right)$. For any two users $i, j \in \mathcal{U}_k$ within the same group, satisfying the preference consistency constraint

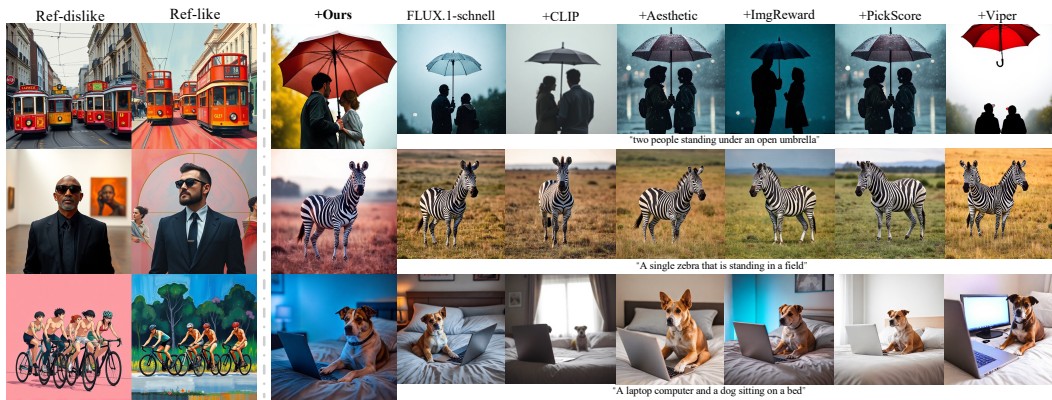

Figure 5: Qualitative comparison of text-to-image generation for three users. Each row shows user preferences (Ref-dislike/like) and generation results from our personalized preference model vs. image-text alignment (CLIP Score), aesthetic quality (Aesthetic Score), general human preference (ImageReward, PickScore), and personalized preference (ViPer) models.

$d(\mathcal{S}_i, \mathcal{S}_j) \leq \rho_k$, there exists a Lipschitz continuous mapping $\phi : \mathcal{S} \rightarrow \mathcal{Q}$ such that: $|\phi(\mathcal{S}_i) - \phi(\mathcal{S}_j)| \leq L_\phi \cdot \rho_k$, where $L_\phi$ is the Lipschitz constant of the mapping $\phi$. Leveraging the Softmax KL Divergence Bound lemma, the similarity of attention responses for users within the same group is constrained below:

$$\mathbb{E}_{\mathcal{K}}\left[\mathrm{KL}(A(\mathcal{Q}_i, \mathcal{K}) \| A(\mathcal{Q}_j, \mathcal{K}))\right] \leq f_{\mathrm{intra}}^{(k)}(\rho_k) \tag{8}$$

where $f_{\mathrm{intra}}^{(k)}$ is a group-specific continuous increasing function. For users from different groups, $i \in \mathcal{U}_k$ and $j \in \mathcal{U}_{l \neq k}$, satisfying the preference separability constraint $d(\mathcal{S}_i, \mathcal{S}_j) \geq \delta_{kl}$, their attention responses are constrained by:

$$\mathbb{E}_{\mathcal{K}}\left[\mathrm{KL}(A(\mathcal{Q}_i, \mathcal{K}) \| A(\mathcal{Q}_j, \mathcal{K}))\right] \geq g_{\mathrm{inter}}^{(kl)}(\delta_{kl}) \tag{9}$$

where $g_{\mathrm{inter}}^{(kl)}$ is an inter-group continuous increasing function.

Following this theoretical intuition, it can be concluded that attention mechanism can adaptively cluster users with different preferences. However, in preference prediction tasks, the context varies for different users, making it difficult for MLLM to formulate groups by adjusting the similarity of attention. Therefore, we introduce additional, shared learnable preference tokens $P_v \in \mathbb{R}^{L_p \times D}$ to provide an extra attention term, where $L_p$ is the number of preference tokens and $D$ is the embedding dimension. This allows group discovery to be achieved by adjusting the attention towards these preference tokens. Given a user's historical sequence $\mathcal{S}$ and a target item $z$, we encode all input content (excluding the target image label token) into a user-specific token sequence $x_u \in \mathbb{R}^{L_e \times D}$. These preference tokens are then concatenated with the user sequence to form the complete input $= [P_v; x_u]$. The Transformer uses attention where the user-specific sequence $x_u$ serves as the Query, and the preference tokens $P_v$ serve as both the Key and the Value, enabling selective activation of relevant preference patterns.

**Mining Similar Users via Attention Mechanism.** To better understand how preference tokens facilitate user similarity modeling and generalization to unseen users, we analyze the learned attention scores $\mathcal{A}$, which capture the interactions between input tokens and preference tokens. Fig. 4 visualizes these interactions, where the same target image is concatenated across users with different reference histories to examine how their preferences are represented. Specifically, Fig. 4 (a) shows User 340 and 649, who exhibit a highly similar pattern of attention across multiple preference tokens, suggesting that they share a common aesthetic inclination. Notably, User 649 is present in the training set, while User 340 is an unseen user. However, the learned preference tokens effectively bridge this gap by encoding shared thematic patterns, such as an affinity for landscapes with dramatic skies, silhouettes, and nightscapes. This observation supports our claim that preference tokens serve as a structured preference representation that captures common aesthetic traits across users, transfers knowledge to unseen users, ensuring that their preferences are accurately inferred without requiring direct memorization of past interactions. In contrast, Fig. 4 (b) illustrates that Users 454 and 700 exhibit distinct attention patterns, revealing that the preference token space does not simply cluster all users together but rather preserves individual differences while leveraging commonalities where applicable. Further details and analysis can be found in Appendix C.

| Model | Aes Score | CLIP Score | ImageReward | HPS Score | PickScore* | IDEFICS | ViPer | **Ours** |
|---|---|---|---|---|---|---|---|---|
| $N_{ref}$ | 0 | 0 | 0 | 0 | 0 | 8 | 8 | 8 |
| accuracy (%) | 49.96 | 53.13 | 55.64 | 56.85 | 57.72 | 50.27 | 55.15 | **61.68** |

\* Trained with the same dataset as our model.

Table 1: Preference classification accuracy on pairwise comparisons.

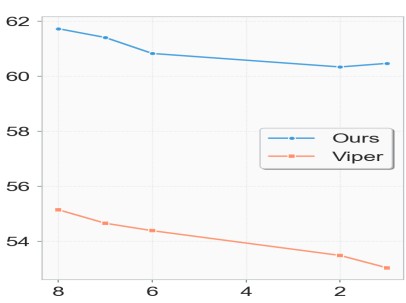

Figure 6: Prediction accuracy with different numbers of images ($N_{ref}$).

Table 2: Accuracy in one-positive-three-negative evaluation setting. We report the top-1 to top-3 accuracy (%).

| Model | $N_{ref}$ | Top-1 Acc | Top-2 Acc | Top-3 Acc |
|---|---|---|---|---|
| Random | 0 | 25.0 | 50.0 | 75.0 |
| Aes Score | 0 | 28.11 | 54.12 | 78.33 |
| CLIP Score | 0 | 30.04 | 55.82 | 76.05 |
| ImageReward | 0 | 31.42 | 58.01 | 78.47 |
| IDEFICS | 8 | 24.40 | 51.88 | 78.33 |
| ViPer | 8 | 31.20 | 56.45 | 78.65 |
| **Ours (w/o $P_v$)** | 8 | 35.72 | 61.64 | 83.44 |
| **Ours** | 8 | **37.47** | **62.85** | **84.74** |

## 4 EXPERIMENTS

### 4.1 USER-SPECIFIC PREFERENCE PREDICTION

**Datasets.** We process Pick-a-Pic v2 dataset (Kirstain et al., 2023), which is collected through real user interactions, to obtain user-specific preference datasets based on user IDs. This large-scale, diverse dataset captures a broad spectrum of aesthetic preferences, making it a strong benchmark for user-specific preference modeling. The processed dataset includes $224, 952$ images and $2, 267$ users in the training set, $1, 707$ images and $89$ users in the validation set, and $2, 234$ images and $70$ users in the test set.

**Implementation Details.** Following the approach of (Salehi et al., 2024), we use IDEFICS2-8B (Laurençon et al., 2024) as our MLLM. We employ a batch size of $64$, training on $8$ A100 (80GB) GPUs with a local batch size of 2 pairs and gradient accumulation over $4$ steps.

**Evaluation Metric.** We evaluate user-specific preference prediction using top-$K$ accuracy, which measures whether the liked image ranks among the top $K$ candidates out of multiple disliked ones.

**Comparison to Other Methods.** In our study, we compare our method with several existing approaches: (1) CLIP (Radford et al., 2021), designed to evaluate generic text-image alignment, (2) LAION Aesthetic Score Predictor (Schuhmann et al., 2021), which evaluates aesthetic quality, (3) ImageReward (Xu et al., 2023), (4) HPS (Wu et al., 2023a) and (5) PickScore (Kirstain et al., 2023), which focus on learning general human preferences and consider relative preferences between images, and (6) ViPer proxy model (Salehi et al., 2024), which is trained on a preference dataset constructed from 5,000 simulated agents representing diverse individual preferences.

**Qualitative User-Specific Preference Prediction.** In Fig. 2, we compare our model to ViPer, PickScore, ImageReward, and Aesthetic Score. Our model effectively aligns with user-specific preferences by distinguishing styles and content according to user reference data. For instance, in Fig. 2 (a), our method accurately captures the user's preference for anime-style imagery with specific attributes such as color, theme, and character features. In Fig. 2 (b), Our method alleviates semantic ambiguity, particularly when handling terms like "grey cat" that encompass multiple visual appearances under a single designation, ensuring that the generated images better reflect the user's intended preferences. More results are in Appendix A.

**Quantitative User-Specific Preference Prediction.** Tab. 2 and Tab. 1 show that our model achieves the highest accuracy, outperforming baselines like ViPer, CLIP, and ImageReward. It performs especially well in settings with multiple disliked images. Generic metrics perform poorly, reflecting the gap between general and personalized preferences. Unlike ViPer, our model captures fine-grained user-specific patterns through contrastive learning and preference tokens, resulting in more accurate and robust predictions.

Figure 7: Human expert evaluation of generated images from different methods on SD1.5-Turbo.

Table 3: Evaluation comparison on preference prediction.

| Method | Top-1 Acc (%) |
|---|---|
| Claude-3.5-Sonnet | 47.96 |
| Human Expert | 57.60 |
| Ours | **60.45** |

Table 4: Ablation study for preference tokens numbers.

| Number of $P_v$ | Top-1 Acc (%) |
|---|---|
| 5 | 61.41 |
| 10 | 61.68 |
| 20 | 61.19 |

Table 5: Quantitative Results. **Bold** and underlined values represent optimal and second-best performance respectively.

| Model | Aesthetic(↑) | CLIP Score(↑) | ImageReward(↑) | PickScore(↑) | HPS Score(↑) | CSD Score(↑) | ViPer (↑) |
|---|---|---|---|---|---|---|---|
| SD1.5-Turbo | 5.81 | **35.44** | **0.69** | 21.92 | 28.04 | 0.32 | 0.21 |
| + ViPer | 5.57 | 34.08 | 0.31 | 21.55 | 27.91 | 0.38 | 0.62 |
| + Ours | **5.99** | 34.45 | 0.60 | **22.28** | **28.49** | **0.42** | **0.84** |

**Number of User Reference Preferences.** As demonstrated in Fig. 6, our method consistently maintains the highest pairwise classification accuracy as the preference sequence length varies. This indicates that our model can effectively preserve accuracy with limited reference data. In contrast, ViPer shows a decline in accuracy as sequence length shortens, highlighting the stability and adaptability of our approach in scenarios with limited user reference.

**Evaluation with Multimodal LLM and Human Experts.** To evaluate real-world performance, we conducted a user study on 200 randomly sampled test cases. Claude-3.5-Sonnet (a state-of-the-art multimodal language model) and ten human experts were asked to infer preferences from reference images and select the preferred image from each pair. As shown in Tab. 3, our model outperforms both Claude-3.5-Sonnet (47.96%) and human experts (57.60%) with a top-1 accuracy of 60.45%. This indicates that our method not only captures clear aesthetic signals but also models subtle user preferences more effectively than humans.

## 4.2 PERSONALIZING GENERATION WITH USER PREFERENCES

**Datasets.** To evaluate our model's ability to learn detailed attribute preferences and guide image generation accordingly, we constructed a diverse dataset. To simulate realistic user preferences, we configured 30 Claude agents (Anthropic, 2024) to represent diverse human preferences across 7 key dimensions: Art styles, Color palette, Composition, Skill level, Detail level, and other aesthetic attributes. To ensure diverse preferences, each agent was configured with unique sets of preferred and dispreferred attributes, maintaining at least 80% Jaccard distance between any pair of agents' preference profiles. For image generation, each agent utilized FLUX.1-schnell (Black Forest Labs, 2024) to generate 10 images aligned with their preferences and 10 images representing their dislikes, resulting in a dataset of 600 images total. Examples of generated images and dataset details are provided in Appendix C.

**Experimental Setup.** Following the ReNO approach (Eyring et al., 2024), which enhances image generation quality by optimizing initial noise during inference using reward model guidance, we generate images guided by our preference model while incorporating both positive and negative user feedback. We first generate initial target images, then utilize the agent preference data as input to our model, which provides reward signals for iteratively optimizing these target images. Specifically, we apply Eq. (6) to obtain the guidance signal, which is subsequently used in an iterative refinement process to enhance target image adherence based on the learned user preferences. We conduct experiments using two generative models: FLUX.1-schnell (Black Forest Labs, 2024) and Stable Diffusion 1.5 Turbo (Sauer et al., 2024b). The images are generated using the same random seeds. Additional experimental details are provided in Appendix C.

Table 6: Clustering evaluation metrics for the ablation study.

| Method | Silhouette Score ($\uparrow$) | Davies-Bouldin Score ($\downarrow$) | Top-1 Acc (%) |
|---|---|---|---|
| Base Loss | 0.596 | 0.812 | 60.47 |
| + $L_{cp}$ | 0.635 | 0.812 | 61.37 |
| + $L_{cp}$ + $P_v$ (Ours) | **0.646** | **0.806** | **61.68** |

**Evaluation Metric.** We employ 7 comprehensive evaluation metrics to assess our model's performance. In addition to 5 standard general-purpose (Aesthetic, CLIP) and human preference metrics (PickScore, ImageReward, HPS), we introduce two specialized metrics: (1) We assess style-following performance using CSD metric (Somepalli et al., 2024) to evaluate how well the generated images maintain consistency with the specified artistic styles and attributes. (2) We utilize the ViPer proxy metric, which predicts user preferences by analyzing reference images.

**Enhancing Image Generation with User Preferences.** As shown in Tab. 5, our method effectively enhances Stable Diffusion 1.5 Turbo (base model) across most metrics by learning richer attribute preferences from user feedback, particularly excelling in personalized preference metrics, and significantly outperforms ViPer across all metrics. This demonstrates that our model can effectively utilize user preferences to refine and guide image generation. Fig. 5 shows generation examples for three different preferences on FLUX.1-schnell with different reward models, where our model successfully extracts personalized preferences and ensures superior image generation results.

**User Study.** To assess the effectiveness of different reward models in guiding personalized image generation, we conducted a user study involving ten human experts. Experts were asked to compare pairs of images generated using different reward models and select the one that better aligns with the reference preferences. As shown in Fig. 7, our method consistently outperforms all baselines, achieving win rates above 79% across all comparisons. This highlights the superiority of our approach in capturing fine-grained user preferences for generation tasks.

## 4.3 ANALYSIS AND ABLATION STUDY

We perform ablation studies and conduct thorough analysis on the processed Pick-a-Pic v2 dataset to investigate the impact of the contrastive preference loss, learnable preference tokens, and the number of preference tokens. To assess whether our model captures structured user preference patterns, we sample 70 users from the test set, each with 16 historical preference images. All users are paired with the same target image, and we extract the final token embedding from the last layer of the MLLM. These embeddings are then clustered using K-means (MacQueen, 1967), and clustering quality is evaluated using Silhouette Score (Rousseeuw, 1987) and Davies-Bouldin Score (Davies & Bouldin, 1979). As shown in Tab. 6, incorporating $L_{cp}$ and $P_v$ significantly improves cluster compactness and separation. This indicates that our method successfully discovers user groupings, aligning with the group structure assumption in our framework. Besides, the full model achieves a top-1 accuracy of 61.68%, representing a cumulative performance gain over the baseline model. Additionally, we perform ablation experiments to analyze the impact of preference token length. As shown in Tab. 4, the results demonstrate that using 10 preference tokens achieves the highest Top-1 accuracy, slightly outperforming configurations with 5 and 20 preference tokens, while the overall differences remain small. This indicates that the selection of preference token quantity exhibits good robustness, effectively enhancing personalized modeling of user preferences within a reasonable range.

## 5 CONCLUSION

In this paper, we propose a novel approach for user-specific preference prediction in generated images by leveraging Multimodal Large Language Models (MLLMs). To address the limitations of existing methods that focus primarily on general human preferences or superficial attributes, we introduce contrastive preference loss and learnable preference tokens. The contrastive preference loss enables the model to distinguish between users' "likes" and "dislikes" more effectively, while the preference tokens capture shared interests across users, enabling both personalization and generalization. Extensive experiments demonstrate that our model outperforms existing methods in preference prediction accuracy, effectively identifying users with similar aesthetic inclinations and providing more precise guidance for personalized content generation.

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
