# OpenReview forum: "Learning User Preferences for Image Generation Models"
_ICLR.cc/2026/Conference — Submitted to ICLR 2026_

### Official Review · Reviewer_Sr5z · 2025-10-27

**Soundness:** 2
**Presentation:** 2
**Contribution:** 2
**Rating:** 4
**Confidence:** 4

**Summary:**

This paper tackles a timely and important problem in AIGC—modeling personalized user preferences for image generation—by highlighting a key limitation of existing methods (e.g., PickScore, ImageReward), which capture only generic human tastes and ignore individual variability and dynamic aesthetic preferences. The problem formulation is well-motivated and has clear practical value for improving user experience in personalized content systems. The proposed approach, leveraging contrastive preference loss and learnable preference tokens, shows conceptual promise in capturing user-specific signals. However, the current submission falls short of acceptance due to insufficient evidence of novelty—many components resemble established techniques in prompt tuning and contrastive learning—and critical gaps in empirical validation.

**Strengths:**

The paper focuses on modeling user preferences in personalized image generation—a research direction of growing importance in the era of AIGC. The authors correctly identify a key limitation of existing methods (e.g., PickScore, ImageReward), which capture only generic human preferences while neglecting individual differences and the dynamic, multi-dimensional nature of aesthetic tastes. This problem formulation is well-targeted and holds direct practical value for enhancing user experience in personalized content generation systems.

**Weaknesses:**

* W1 : The core modeling assumption (Assumption 1) posits that users naturally form groups with “intra-group homogeneity and inter-group heterogeneity.” However, this assumption is not empirically validated on real user behavior data. Although Table 6 shows improved clustering metrics, these improvements may stem from the model’s inductive biases rather than reflecting genuine user structural patterns. The authors are encouraged to provide an actual clustering analysis of user preferences in the Pick-a-Pic v2 dataset—e.g., through visualization or statistical tests—to substantiate the plausibility of this assumption.

* W2:  Although Tables 4 and 6 present some ablation results, they lack in-depth analysis of critical design choices. For instance: (1). Is the specific formulation of the contrastive loss (L_CP)—including margin design and negative sampling strategy—optimal? (2). While preference tokens are integrated via attention with user historical interactions, it remains unproven whether they truly enable “cross-user knowledge transfer” (e.g., is the example in Figure 4 statistically significant?). (3).The authors are advised to conduct quantitative analyses of preference token activation patterns—such as mapping token activations to user clusters—to empirically validate the underlying design rationale.

**Questions:**

See weakness.

---

### Official Review · Reviewer_f7xP · 2025-10-29

**Soundness:** 3
**Presentation:** 3
**Contribution:** 3
**Rating:** 8
**Confidence:** 2

**Summary:**

This paper solves the problem of personalized preference prediction for image generation. It purposes two ideas: 1. a contrastive preference losses that "enhance the model's ability to differentiate between like and dislike predictions by emphasizing their relative rankings." 2. learnable preference tokens that capture shared aesthetic traits across users. The result of the method proposed by the authors is promising. The authors not only include the quantitive results, but also include qualitative results with respect to human experts.

**Strengths:**

This paper is well-written. The motivation is clear and the method is presented in the diagram nicely. The experiments are also thorough. The methods achieve SOTA performance on various large, real-world dataset. The paper also includes quantitative and quantitative analysis as well as human expert evaluation.

**Weaknesses:**

I think the author could make some of the graphs look nicer. For example, fonts in Figure 6 and Figure 7 seem like to be stretched in some way. I also wonder if it is possible to have error bars for the numbers in the table (for example, in table 1).

**Questions:**

The authors mention on line 205 that L() denotes a classification loss function. Is it a cross-entropy loss?

---

### Official Review · Reviewer_YQvV · 2025-10-30

**Soundness:** 2
**Presentation:** 3
**Contribution:** 2
**Rating:** 4
**Confidence:** 4

**Summary:**

This paper presents an approach for learning user preferences for image generation models. The proposed approach is built upon Multimodal Large Language Models, equipped with contrastive preference loss and preference tokens to learn personalized user preferences from historical interactions. The contrastive preference loss is designed to effectively distinguish between user “likes” and “dislikes”, while the learnable preference tokens capture shared interest representations among existing users. Experiments demonstrate the effectiveness of the proposed method.

**Strengths:**

A MLLM-based contrastive learning framework that enables the model to learn discriminative features from users’ liked and disliked data is introduced.

Learnable preference tokens are utilized to capture shared interests among users, allowing the model to generalize better across users with similar tastes.

Experimental results demonstrate that our model outperforms existing methods in preference recognition accuracy.

The paper is well written and easy to read.

**Weaknesses:**

My main concern is that this work has relatively limited novelty. First of all, learning user preferences with MLLM is now a well-adopted paradigm. Besides, learning discriminative features from users’ liked and disliked data via contrastive learning is common strategy in both the research field of image aesthetic assessment and recommendation systems. In my opinion, the main difficulty lies in how to collect large-scale data for training. However, this work does not provide feasible solution to this problem.

The experiments are less convincing. The authors are suggested to analyze the impact of different MLLM, and the related parameters. Many recent methods are not compared, including CycleReward [ICCV2025], UnifiedReward [arXiv2025], UnifiedReward-Think [NeurIPS2025], and LLaVA-Reward [ICCV2025].

The paper lacks evaluation on generalization of the proposed method to images from unseen domain. In addition, a thorough analysis on computational cost and efficiency is missed.

It would be better to provide visual examples to intuitively validate the effectiveness of the contrastive preference loss, learnable preference tokens, and the number of preference tokens.

**Questions:**

Is it possible to make the trained model to output the preference scores?

It would be better to analyze what is the key visual element that determine the learned user preference. It appears to me that color tone of the image is the key element, as indicated by visual results in Figure 5.

There are some grammar errors. For instance, for the sentence of 'PickScore (Kirstain et al., 2023), which focus on learning general human preferences', 'focus' should be 'focuses'.

---

### Official Review · Reviewer_9G7c · 2025-10-31

**Soundness:** 3
**Presentation:** 2
**Contribution:** 2
**Rating:** 2
**Confidence:** 4

**Summary:**

The paper proposes a method for learning a user’s visual preferences based on their liked and disliked images. The approach trains a MLLM using a contrastive loss. To capture shared interests among users, learnable preference tokens are appended to the input representations. The paper shows supportive results demonstrating the effectiveness of their model in preference prediction and its utility for personalized image generation.

**Strengths:**

The paper was well written and easy to follow.

**Weaknesses:**

- The overall goal of the paper is unclear. It is not evident whether the focus lies in developing a method for predicting user preferences, or in improving personalized image generation.
    - If the main goal is personalized generation, it remains unclear why personalization via a preference prediction model is preferred. Moreover, there are more recent and relevant baselines [1,2] beyond ViPer, and simpler methods that learn personalized prompts through e.g., textual inversion.

        [1] Kim et al., Draw Your Mind: Personalized Generation via Condition-Level Modeling in Text-to-Image Diffusion Models

        [2] Chen et al., Tailored Visions: Enhancing Text-to-Image Generation with Personalized Prompt Rewriting

    - Conversely, if the goal is to build a preference prediction model, potential downstream applications should be discussed. E.g., could this model be used to recommend images, or trained personalized models?
    - The paper frequently references user groups, yet it is not clear what the benefits of clustering users are. While grouping may emerge naturally from the learned preference tokens, it appears to serve as a visualization tool rather than for additional insights.
- The use of preference tokens appears to be a central design component and contribution. Additional analysis could strengthen the paper:
    - Is it possible to visualize what each token represents, for instance by projecting them onto interpretable hard tokens?
    - The number of preference tokens seems to be an important hyperparameter intended to capture representative user attributes. However, the reported results suggest limited variance across different token counts.
- The paper claims that, unlike previous methods which focus on superficial features such as color or style, the proposed approach captures deeper semantic preferences. However, the qualitative examples (e.g., Figure 5) seem to suggest that the model still primarily reflects color and stylistic preferences. It would be useful to demonstrate whether the method can indeed learn more abstract or semantic user preferences e.g., cultural preferences.

**Questions:**

- The liked (or disliked) images appear to be quite similar. Can the method model diverse preferences within a single user (e.g., multiple distinct aesthetic or thematic interests)?
- Does the method learn semantic preferences beyond features such as color or style?
- Was the same base image generation model used for both ViPer (or other baselines) and the proposed method?

---

### Meta-Review · Area_Chair_Q5tn · 2026-01-09

**Summary:**

This paper highlighted the challenge of learning personalized user preferences for image generation models by proposing an approach built upon Multimodal Large Language Models (MLLMs) that incorporates contrastive preference loss and learnable preference tokens to capture individual tastes from historical interactions. This paper has clear limitations regarding the novelty and technical depth.

**Reviewer Concerns:**

Most concerns

**Reviewer Scores:**

Unlike

---

### Decision · Program_Chairs · 2026-01-26

Reject